# Through the Lens: Youth Experiences with Cancer in Rural Appalachian Kentucky Using Photovoice

**DOI:** 10.3390/ijerph19010205

**Published:** 2021-12-25

**Authors:** Katie Gaines, Courtney Martin, Chris Prichard, Nathan L. Vanderford

**Affiliations:** 1Markey Cancer Center, University of Kentucky, Lexington, KY 40506, USA; katie.gaines7@uky.edu (K.G.); courtney.martin@uky.edu (C.M.); chris.prichard@uky.edu (C.P.); 2Department of Toxicology and Cancer Biology, College of Medicine, University of Kentucky, Lexington, KY 40506, USA

**Keywords:** Appalachia, rural, cancer, cancer disparities, youth, photovoice

## Abstract

Rural Appalachian Kentucky experiences disproportionately high cancer incidence and mortality rates. This cancer burden is due to social determinants of health and cultural factors prominent in the region. The firsthand experiences of community members—especially young people—can highlight these factors and identify areas for improvement. The purpose of this study was to encourage Appalachian Kentucky youth to consider determinants of cancer and visualize the effects that cancer has on their families or communities by asking them to take photographs of cancer-related objects around them. Content analysis was performed on 238 photographs submitted by 25 students, and photographs were organized into themes, subthemes, and subtopics. The six themes that emerged were risk factors and exposures, marketing, awareness and support, health care, experiences, and metaphorical representations. Many of the submitted photographs aligned with cultural, environmental and/or situational factors prevalent in Appalachian Kentucky. Of the submitted photographs, 54 were displayed as an installment in two Kentucky art galleries. Viewer comments at the exhibitions demonstrated that young community members can educate and motivate change in those around them. Ultimately, this project demonstrates that young community members can recognize cancer-related issues around them and connect personal experiences back to the larger Appalachian Kentucky cancer disparity while also having an impact on other community members.

## 1. Introduction

Cancer is the second leading cause of death in the United States (US) annually, behind heart disease, and although cancer death rates have decreased 31% in the past thirty years, some areas of the country still struggle disproportionately from the disease [1]. In recent years, Kentucky has consistently ranked first in overall cancer incidence and mortality rates within the US, and, importantly, overall cancer incidence and mortality rates are significantly higher in the Appalachian region of the state [1,2,3,4]. Appalachian Kentucky is widely known for its culture and history of tobacco growth and use. In fact, Appalachian Kentucky has much higher lung cancer incidence and mortality rates compared to the rest of the US due to increased smoking rates [1,4]. Social and ecological factors have also contributed to the cancer burden in Appalachian Kentucky: lower income and education levels, barriers to access, lack of screening and immunization, increased obesity rates, environmental exposures, and unsupportive health policies have fostered a patient population which is more susceptible to various types of cancer [4]. Further, many people living in Appalachia (and especially in Appalachian Kentucky) hold negative perceptions regarding cancer prevention and diagnoses such as “everything causes cancer,” “there are too many cancer prevention recommendations to follow,” as well as equating cancer to death [2,5]. Appalachians are also more likely to avoid seeing a physician and to believe they (Appalachians) can tell if they have cancer [5].

Given the high rates of cancer and socioeconomic disparities facing the region, there is an opportunity to target and engage youth from the area in programs that can provide enhanced educational opportunities including those that prepare them for cancer-related careers. By reaching out to youth who have ties to the region and/or goals to serve the region, engagement efforts can invest in students to become resources for their own underserved communities. The Appalachian Career Training in Oncology (ACTION) Program is an initiative by the Markey Cancer Center at the University of Kentucky which provides high school and undergraduate students from Appalachian Kentucky an opportunity to participate in cancer research, clinical shadowing, education and career development, and community outreach and engagement activities. The goal of this program is to prepare the next generation of Appalachian Kentucky health care providers, researchers, and education specialists and, through community engagement, increase cancer awareness and literacy levels in the region. Ultimately, ACTION aims to reduce the cancer disparities in Appalachian Kentucky. Students who are in the ACTION program have a higher level of cancer education than most of their community members, equipping these participants with the knowledge to make connections between their own experiences and the cancer disparities faced by Appalachian Kentucky [6]. In a book of essays from ACTION students, one student writes about his personal experience with cancer through the diagnosis of a close family member, saying “until that moment, cancer had felt like a toll on the turnpike of life, but I never thought it could be so expensive” [7]. Hearing and seeing the firsthand experiences of young Appalachian community members can be a powerful motivator for others to inspire social change.

A community-based participatory research method known as photovoice was utilized in this study. Photovoice asks participants to take photographs and provide narratives which chronicle their everyday, personal experiences. The photovoice method provides a means for participants to record and reflect on their situations or surroundings, promote conversation and understanding of personal and/or community issues, and engage community and/or policy stakeholders in dialogue related to situational or community improvements. Photovoice has been applied as a health promotion strategy [8,9,10]. Further, Cardarelli et al. suggested that photovoice is an especially useful method for researching vulnerable populations which have been either misrepresented or not represented at all by the media [11]. Cardarelli and colleagues argue that Appalachia has historically been stereotyped in the media, making photovoice an advantageous technique for the unique population [11].

The current study aims to engage youth by prompting them to visualize the causes and consequences of cancer in their communities and to inspire outreach efforts which may begin to address these cancer causes and consequences. Visual and written narratives can also personalize the disparities Appalachian Kentuckians experience rather than using numbers and statistics, allowing for a deeper understanding of how these communities experience health disparities at the individual level. To our knowledge, this study is the first to utilize photovoice to analyze the impact of cancer on Appalachian youth.

## 2. Materials and Methods

Participants in this study included high school and undergraduate student members of the ACTION program. Participants were asked to take photographs of objects in their communities which reminded them of cancer and include a short caption for each photograph. Participants were instructed to use their cell phone cameras to take the photographs and they were not provided with cameras by the researchers. This allowed for a convenient and organic photovoice experience since most individuals carry their cell phone with them at all times. This allowed participants to document objects during their everyday routines without the need for bulky camera equipment. Participants did not receive any specific or formal photovoice or photography training related to this study and they were not compensated specifically for participation in the project.

Photographs were collected from the participants electronically and uploaded to an online file storage and sharing system. In total, 25 students submitted photographs for this study. Demographic information about the participants can be found in Table 1. Each student submitted between eight and eleven photographs, for a total of 238 photographs. Photograph collection began in the fall of 2019 (approximately September 2019), and submissions continued through spring of 2021 (approximately April 2021). Participants were not given access to view all of the photographs during the data collection period. This was done intentionally to avoid participants being biased or influenced by each other’s work.

Qualitative content analysis was used to identify themes among the photographs. The photographs were grouped into broad themes, then analyzed further to identify potential subthemes and subtopics. Only one theme, risk factors and exposures, was large enough to identify subtopics. Four photos were unable to fit into the themes or subthemes identified by researchers and were therefore excluded from the analysis. Captions provided by the students were considered by researchers to provide context for the photographs. Photographs were qualitatively analyzed by the primary coder, KG, then the analysis was reviewed by a secondary coder, NLV, and a third coder, CM, reviewed the code and aided in consensus building. Modifications were made until a consensus was reached.

In collaboration with the University of Kentucky Arts in HealthCare program, 54 photographs were chosen for display at the Pam Miller Downtown Arts Center in Lexington, Kentucky, and the Kentucky Folk Art Center in Morehead, Kentucky. The Lexington exhibit was open 10 September 2021, through 30 October 2021, while the Morehead exhibit was open from 1 October 2021, to 9 December 2021. The displayed photographs were selected in alignment with the dominant photograph themes. Selection was also based on artistic expression, photograph quality, perceived ability to resonate with the community, and budget constraints. Comments were solicited from visitors to the exhibits to obtain feedback and impressions about the photographs. Thematically representative comments are reported. The exhibits were open to the public during extended periods of time throughout the day. An attendant was not always present and a count of the number of visitors was therefore not taken.

This content analysis study was approved by the University of Kentucky Institutional Review Board (#60277). Student engagement with the ACTION program itself was also approved by the Institutional Review Board (#44637).

## 3. Results

Content analysis of the photographs yielded six major cancer-related themes: risk factors and exposures, marketing, awareness and support, health care, experiences, and metaphorical representations. Some of the photographs that were categorized into these themes were further grouped into subthemes and subtopics.

### 3.1. Theme 1: Risk Factors and Exposures

The largest theme that emerged, risk factors and exposures, was organized into five subthemes and 14 subtopics (Table 2). Participants submitted photographs displaying several exposures in their homes and communities which they perceived as risk factors for cancer. These exposures included products sold for consumption such as fast food, alcohol, and various forms of tobacco (Figure 1). Participants also identified chemicals such as pesticides, household cleaners, paint, and gasoline as potential risk factors that were commonly used around their homes and communities. Tobacco farming and coal mining were depicted as well, with accompanying captions describing their cultural significance and their contribution to increased cancer risk in rural communities. Other environmental exposures such as UV rays, fumes, radiation, and water pollution were identified in photographs in this theme.

### 3.2. Theme 2: Marketing

Distinct from the photographs of risk factors themselves were photographs of various marketing techniques which promoted the sale of these risk factors (Figure 2). Billboards and window signs boasted cheap tobacco products and emphasized the location of stores which sold tobacco. Photographs included tobacco “barns,” which conveniently provide consumers with the option to buy tobacco products at a drive-through window.

### 3.3. Theme 3: Awareness and Support

Participants submitted photographs of cancer awareness and support initiatives which took place in their communities (Figure 3). While some initiatives were planned and organized in support of specific community members who had been diagnosed with cancer, others were simply traditions or annual events. Photographs included shirts, signs, and vehicles with messages about cancer, images of awareness ribbons, and names of affected community members.

### 3.4. Theme 4: Health Care

Participants indicated that various health care resources in their communities, or lack thereof, reminded them of cancer (Figure 4). Cancer treatment facilities were often small treatment centers serving as remote branches of larger health care organizations. Also submitted were photographs of objects which reminded participants of the lack of access community members have to health care resources in rural areas. One photograph depicted a trailer, which the participant explained in his caption represented communities of lower socioeconomic status, which have higher cancer mortality rates. Another photograph showed a mountain range in rural Kentucky, which the participant described as a geographical barrier to care.

### 3.5. Theme 5: Experiences

Photographs also captured objects that reminded the participants of their personal experiences with cancer (Figure 5). Photographs in this theme included belongings of loved ones who had been diagnosed with cancer or who had passed away from the disease. According to the submitted captions, items such as hats, scarves, shoes, and jewelry were representative of friends and family members of the participants who had an experience with cancer. Four photos depicted memorials and graves of those who had lost their lives to cancer. Five of the photos captured medications, test results, and assistive devices of loved ones being treated for cancer. Nine photos captured the loved ones themselves, typically taken at a time before the cancer diagnosis, during treatment, or while spending time with family members and friends.

### 3.6. Theme 6: Metaphorical Representations

Five of the submitted photographs included miscellaneous objects which participants metaphorically linked to cancer through their captions (Figure 6). These included a butterfly signaling the need for change in cancer treatment in the community, the roots of a tree reminiscent of the way cancer “takes root” in an affected person, and weeds representing how cancer can metastasize in the body.

### 3.7. Community Response

At the time this manuscript was written, 54 photographs were being displayed at two art galleries: the Pam Miller Downtown Arts Center in Lexington, Kentucky, and the Kentucky Folk Art Center in Morehead, Kentucky [12]. It is noteworthy to mention that while the Lexington exhibit took place in a more urban area, the Morehead exhibit was located in a rural, Appalachian Kentucky community similar to where the photographs were taken. Thus, viewers from various types of populations were exposed to the messages expressed by these photographs. Visitors to the exhibition of these photographs were given the opportunity to leave written comments and impressions. One viewer expressed that they had learned from the exhibit, stating they “did not know Kentucky has highest incidence rate of cancer”. Another put what they learned simply, commenting “smoke = bad”. Other visitors expressed the emotional impact of the exhibit, saying it “points to the consequences of our actions” and “I hope this helps change some mindsets, maybe gets someone to quit smoking”. One viewer commented “I have been in Kentucky for about 7 days, and I recognize two of the things this installation has documented that increases the risk of cancer. Poor nutrition and smoking!” A University of Kentucky professor even stopped by the exhibit, utilizing the comment book to remark “Powerful show. Hurts like it should”. At the time of this writing, 34 unique comments had been left by visitors to the public exhibits, demonstrating the impact this photovoice project has had on the community. The comments included above are thematically representative of the majority of comments left by visitors.

## 4. Discussion

This study utilized a photovoice approach, asking young participants from rural areas within Appalachian Kentucky to capture photographs of objects in their communities that remind them of cancer. This study aimed to inspire these participants to contemplate contributing factors and the effects of cancer in their communities as well as to consider potential outreach efforts which might be helpful in confronting the cancer disparities that rural Kentucky communities experience. We found that most photos submitted by participants captured known cancer risk factors such as tobacco, diet, environmental exposures, and chemicals. Cultural factors such as diet and community history of coal mining and tobacco farming were also included in the submitted photos, suggesting that culture tends to play a strong role in how rural Kentucky communities experience cancer. This study also demonstrated the capability of youth in multiple ways. Young students conducted research in their own communities and were able to make connections between the cancer disparities facing the region and their own experiences. Impressions from viewers at the exhibition of the photos taken by these young participants also demonstrated that young individuals are apt to educate others around them and be agents of change in their communities.

This project is not the first to utilize photovoice to assess health experiences in Appalachian youth. A 2019 study by Cardarelli et al. asked students to take photographs which visualized contributors and consequences of respiratory disease in Appalachian communities [11]. Specifically, regarding cancer, photovoice projects have also been conducted to assess the experiences of young siblings of children diagnosed with cancer, as well as the perceptions of young cancer patients themselves [11,13,14,15]. Photovoice has also been used to engage youth in rural communities regarding increased obesity rates [16]. The method has also been employed to facilitate discussion surrounding substance abuse and mental illness in communities [17,18]. Unlike some of these studies, our study did not host sessions to facilitate discussion about the photographs. Participants also did not view or discuss each other’s photographs prior to the final photovoice submissions, which we believe allowed for greater individuality and resulted in a wider variety of photographs. In doing this, we wanted to avoid students being biased by seeing each other’s photographs. However, participants have been given the opportunity to discuss the cancer disparities in Appalachian Kentucky in other ACTION program events. Ultimately, to our knowledge, this is the first study to utilize photovoice to engage Appalachian youth regarding their experiences with cancer. This work has significant potential given that community-based participatory research can address and perhaps ameliorate cancer disparities [19,20,21].

This study may be limited by its small sample size, as there were 25 participants. However, many other photovoice studies have less than 20 participants [10,11,13,14,15,22,23,24]. Further, our 25 participants submitted 238 photographs which were analyzed leading to identification of six main themes. Additionally, this study may not be generalizable across different areas of the country as participants were limited to those from rural Appalachian Kentucky. Findings may also differ across various racial or ethnic groups or based on different situational, environmental or cultural factors. For example, almost all participants in this study were white, which does not account for the experiences of other people groups or subcultures. Nonetheless, findings may be similar if this study were conducted among other mostly white individuals in other rural areas, particularly in the southern US, who also experience cancer disparities. Additionally, the qualitative analysis for this study was conducted manually, which may inherently introduce analysis bias. However, this bias was controlled for by utilizing multiple reviewers. The participant sample also had the potential for bias, in that, all participants were student members of the ACTION program. This program provides education about cancer risk factors and effects, meaning students were knowledgeable about cancer prior to participating in this study. This exposure to cancer education may have influenced the types of photographs that participants submitted, especially since even a brief cancer education intervention can enhance Appalachian Kentucky students’ understanding of cancer [25,26]. The exact community impact of this work through our exhibits is also unknown. The exhibits were made open to the public without an attendant present at all times to count the number of visitors. However, the comment book was made openly available to visitors, and they were encouraged to leave comments if they wanted to do so. Finally, this study did not specifically seek to change public health policy, as many photovoice studies do. The focus of this photovoice project was to engage youth in a cancer-related community-based participatory research project and influence public opinion and raise awareness of the rural Appalachian Kentucky cancer disparity rather than target a specific policy or policymakers. However, when the opportunity arises, our future work with these photographs could aim to effect change in certain policies or regulations within Kentucky in general, within the Appalachian Kentucky region, and/or in specific Appalachian Kentucky communities.

Overall, the ability of youth to recognize causes and consequences of health disparities in their own communities and to educate others suggests that younger community members have the potential to serve as agents of change in the health behaviors of those around them. This suggests that youth can inspire interventional efforts for various health conditions, not only cancer. Future research may find value in targeting other health conditions like heart disease and diabetes, which Appalachian Kentucky also suffers from disproportionately according to the Appalachian Regional Commission [27]. Further photovoice study frameworks may aim to impact disease prevention or public health policy within the community as well.

## 5. Conclusions

Using photovoice, Appalachian youth were able to connect objects in their own communities to the cancer disparities in rural Appalachian Kentucky and motivate changes which may serve as a valuable method to address these disparities. Participants submitted photographs capturing cancer risk factors and exposures, marketing advertisements for those risk factors, awareness and support initiatives, health care issues, personal experiences with cancer, and metaphorical representations of cancer. Most submitted photographs visualized known risk factors and exposures linked to cancer, while also highlighting references to cultural factors such as tobacco farming, coal mining, and traditional diet which imply that community members acknowledge the large role that sociocultural factors play in Appalachian Kentucky cancer morbidity and mortality. This study also demonstrates that youth are able make connections between their own experiences and health disparities at large, and through comments from the exhibition of the photographs, youth community members appear to be able to serve as agents of change to those around them. This means young people can influence others around them by, for example, promoting community members to consider adverse health behaviors and policy interventions which may directly impact the disparities their communities face.

## Figures and Tables

**Figure 1 ijerph-19-00205-f001:**
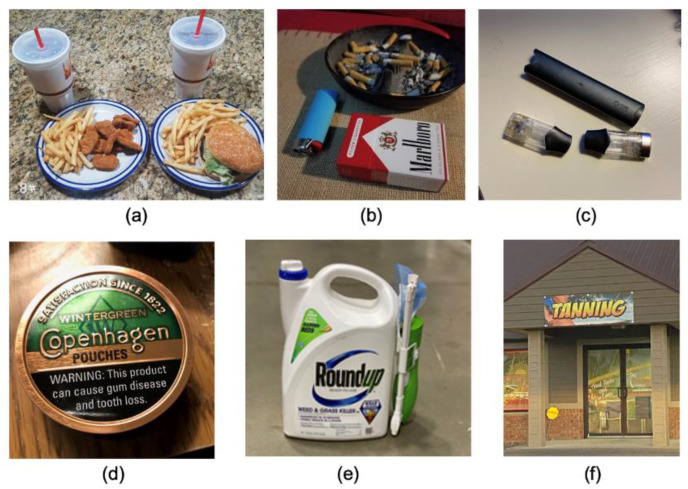
Risk factors and exposures: Diet, tobacco, and chemicals. Student-submitted captions: (**a**) “Continually eating an unhealthy diet can contribute to obesity, which can lead to cancer”; (**b**) “This picture represents cancer to me because smoking is one of the main causes [of] cancer”; (**c**) “A high school student’s vaping device, confiscated by his mother”; (**d**) “Can of Copenhagen that belongs to a 78-year-old male from Eastern Kentucky”; (**e**) “Roundup is the most widely used herbicide in the world. Roundup contains glyphosate. Large amounts of glyphosate can lead to Non-Hodgkins lymphoma”; (**f**) “Devices like tanning beds and sun lamps can emit higher amounts of ultraviolet radiation than the sun, including both UVA and UVB radiation. UV radiation of any type increases your cancer risk, and the more you get, the higher your risk”.

**Figure 2 ijerph-19-00205-f002:**
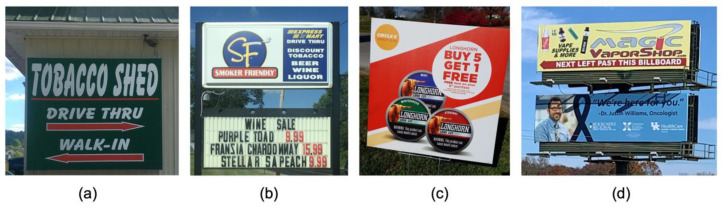
Marketing: Advertisements for tobacco products and tobacco barns. Student-submitted captions: (**a**) “Cancerous convenience”; (**b**) “Cancer friendly?”; (**c**) “This display sign at a local gas station seems to be encouraging people to purchase their smokeless tobacco. The labels explicitly state that use of this product may cause mouth cancer”; (**d**) “Irony in marketing”.

**Figure 3 ijerph-19-00205-f003:**
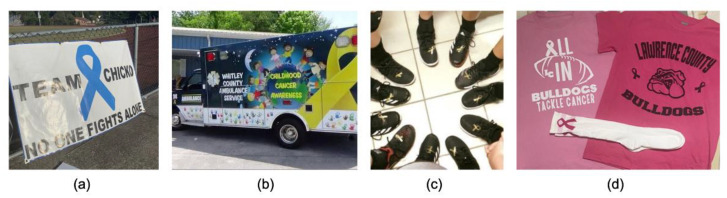
Awareness and support: Community events to raise cancer awareness. Student-submitted captions: (**a**) “A banner hung on the fence of the high school football field to honor a beloved coach and father of five during his battle with cancer. The banner has stayed up long after his passing”; (**b**) “One of Whitley County’s prized ambulances raising awareness for childhood cancer”; (**c**) “Whitley County’s volleyball team wears ribbons in support of a senior student being diagnosed with testicular cancer”; (**d**) “These three items have been worn by my cheerleading team at football games. Cheerleaders are noticed by many people on Friday nights, so wearing items like these will surely raise awareness in the community. Raising awareness is crucial to lowering the number of cancer cases in Kentucky”.

**Figure 4 ijerph-19-00205-f004:**
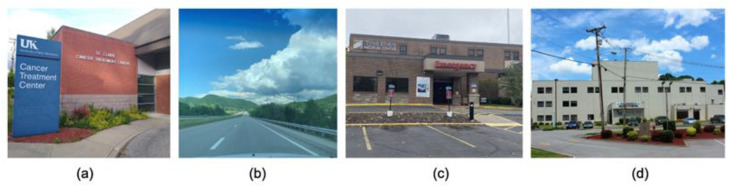
Health care: Health care resources and barriers. Student-submitted captions: (**a**) “A cancer treatment center in Morehead, KY”; (**b**) “While driving home from Lexington, the mountains in Stanton, Kentucky, are the first sign I’m getting close. These mountains, while beautiful, represent barriers that many face while getting cancer treatment in eastern Kentucky”; (**c**) “Many Kentuckians avoid seeing a doctor, even if they are experiencing symptoms of cancer. When less people are screened for cancer, less cases are caught before it’s too late”; (**d**) “This little hospital in Paintsville is where many may have their first interaction with cancer, whether it is getting a preventative procedure, a surgery, or visiting a sick friend or family member”.

**Figure 5 ijerph-19-00205-f005:**
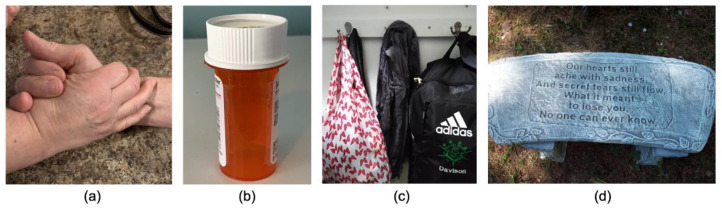
Experiences: Loved ones suffering from cancer and objects reminiscent of them. Student-submitted captions: (**a**) “This is a picture of my grandparents holding hands. When my Mamaw was diagnosed with breast cancer, my Papaw dedicated all of himself to helping her as much as he could. He exhibits the gentleness and love of caretakers, who help their loved ones when they need it most”; (**b**) “This is a picture of a medicine bottle. When a person goes through cancer, they must take several types of medication”; (**c**) “A scarf my mother wore on her head during her chemotherapy treatments, forgotten in the closet”; (**d**) “My beloved Papaw was 62 when he lost his battle with cancer. I was three years old”.

**Figure 6 ijerph-19-00205-f006:**
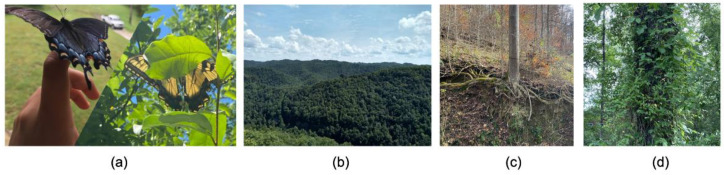
Metaphorical Representations: Objects figuratively representing cancer. Student-submitted captions: (**a**) “A photo of a butterfly seems like a weird way to express what cancer means to you, but do you know what a butterfly resembles? A butterfly resembles change. Kentucky needs change, we need a breakthrough. Every time I think of My Old Kentucky Home, I think of what it means to me, and how important it is that we see a decrease in cancer cases. I believe we can do it, I believe we will see a change”; (**b**) “Appalachia, a beautiful corner of our state, is one of the biggest cancer hotspots in the United States”; (**c**) “This may seem like an unconventional way to show what cancer means to me. But when I look at this picture, I see something taking root in the surrounding area. I see something embedding itself into the pre-existing life around it. I see something trying to survive off of neighboring life. Just like how cancer does. Cancer takes root in a person, and it is grueling to get rid of once it is lodged in place—just like the entangled roots of this tree”; (**d**) “The way cancer can metastasize in someone’s body reminds me of how vines and weeds can take over forests if left uncared for”.

**Table 1 ijerph-19-00205-t001:** Participant Demographics.

Parameter	Frequency (*n*)	Percent (%)
Academic Level		
Undergraduates	2	8
High school students	23	92
Gender		
Male	7	28
Female	18	72
Race/Ethnicity		
American Indian/Alaska Native	0	0
African American/Black	0	0
Asian	0	0
More than one race	2	8
White	23	92
Hispanic or Latino	2	8
Not Hispanic or Latino	23	92
Disparity Status		
1st generation	9	36
Low income	10	40
Rural resident	25	100

**Table 2 ijerph-19-00205-t002:** Theme 1: Risk Factors and Exposures.

Subtheme	Subtopic
Tobacco Farming	None
Tobacco	Vape, Smokeless Tobacco, Cigarettes
Environmental Exposures	Water Pollution, UV Exposure, Radiation, Fumes, Coal
Diet	Food, Alcohol
Chemicals	Pesticides/Weed killer, Paint, Household Cleaners, Gasoline/Benzene

## Data Availability

The photographs and captions analyzed for this study are being made available at https://www.instagram.com/cancer_in_appalachia/ (accessed on 24 December 2021).

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
