# Peer review of "Through the Lens: Youth Experiences with Cancer in Rural Appalachian Kentucky Using Photovoice"

_ijerph, 2021, doi:10.3390/ijerph19010205_

Round 1

Reviewer 1 Report

This is a well written paper that uses photovoice to identify social and individual factors associated with cancer in rural Appalachian Kentucky. A few minor comments are suggested to enhance the reproducibility of this work and to link it to the broader body of related work. Overall this is an excellent contribution to the field.

Abstract. None. Looks great.

Intro. Looks great.

Methods.

  • Did you model your photovoice approach off an existing framework? If not, why? If yes, what did you alter (for example it appears everything happened remotely, this is an innovation and has the potential to make the method more accessible to some – but also perhaps to exclude others).
  • Did students use their own phones/cameras to take pictures? Did they receive an incentive for participating?
  • Was any guidance provided for students in preparing their captions? (These are all questions b/c they are common decision points in photovoice projects)
  • Why didn’t you allow students to see the other photos? In the prior photovoice studies I’ve done, group reflection on the photos has helped seed thematic identification.
    • Note I see you address this in the discussion on page 8. Did you gather any feedback from the students about this change – i.e., would they have wanted to see other’s photos and if so, why?
  • [Important] Page 3, line 94-101: can you clarify which of the authors were involved in which coding steps? (Note I initially wondered who/how many had been involved in this step, but this info appears at the end of the paragraph).
  • [Important] When were the photos collected? When were they analyzed?
  • Page 3, line 106: Did dominant themes not factor into photo selection? This seems like a missed opportunity if it was not considered as part of what informed the public display.
  • See comment below re: additional information about community events and “themes” from these events/feedback

Results

  • Formatting of Theme 2 is not the same as others (font, level). May be helpful to state the dominant themes in your opening paragraph to help orient readers.
  • Do you know how many individuals from the public saw the photos? This would be nice info to add if available. Note that the sentence on page 7 line 235 and 236 reads like methods. It would be helpful to put info in the methods about how your team gathered and reviewed feedback (i.e., what was the research process to decide which comments you shared in the results and why)?
  • I’d edit to de-identify this individual on page 7 line 243-244 (I’m guessing some would be able to guess who it is based on the level of detail provided). I’d take out the poet laureate.
    • A University of Kentucky professor and former Poet Laureate of Kentucky even stopped by the exhibit, utilizing the comment book to remark “Powerful show. Hurts like it should.”

Discussion

  • I don’t think your sample size is small, for a photovoice project. I would encourage you to add some citations to this limitations paragraph to indicate that your sample is relatively large compared to other studies and (if true) captures the sample demographics your team was searching for. Don’t allow traditional “quantitative” research standards pollute what is considered rigorous in photovoice/qualitative methods.
  • Key limitation is that although you state findings may be relevant to other areas in the south (page 8, line 281), Table 1 indicates that there are no black participants in your sample. I think this may be something you want to edit or highlight as an opportunity for future work. I would be cautious here as your results capture white Appalachian views – but perhaps not some of the barriers/factors that might impact people of color in other southern states. If you disagree, make it more clear as to why.

References

  • There are only 11 references in this manuscript. I’m not sure if that’s what’s normal for this journal – but it may be helpful to the authors to place their work in light of other photovoice studies or qualitative research studies that explore youth views toward cancer…

Reviewer 2 Report

The study conducted by Gains et al. is an important example of how education and public awareness of science can impact in people perception of diseases in general, and particularly of cancer.

I consider it is important to include studies of this type in the journal, since a change is needed in the communities from the new generations to achieve an improvement in public health in future years.

I find the study very interesting. However, it could be useful with a larger number of participants.

Beyond the small sample size, and that the participants belong to the ACTION program, could the authors recognize other limitations of the study?

I believe that the authors also could investigate what was the influence of the photographs in participants’ families.

Moreover, the authors could argue about the importance of the involving of communities in preventions of diseases and the impact of this study in the public health politics.

In the Community response results, it is important to know the scope of the exhibition, for example, how many people received the message, and how many responded in the guest book. Since only what some visitors wrote is mentioned. This later could help to understand the effect that just 25 people can obtain in a community, and how these programs would be implemented in larger populations.

I coudn't find the supplementary materials.

Reviewer 3 Report

This important article describes and summarizes a local educational program for youth and it is very interesting to read. It can be published as a special issue of a journal.

Small sample size is a huge limitation.
